# CircRNAs Related to Breast Muscle Development and Their Interaction Regulatory Network in Gushi Chicken

**DOI:** 10.3390/genes13111974

**Published:** 2022-10-29

**Authors:** Pengtao Yuan, Yinli Zhao, Hongtai Li, Shuaihao Li, Shengxin Fan, Bin Zhai, Yuanfang Li, Ruili Han, Xiaojun Liu, Yadong Tian, Xiangtao Kang, Yanhua Zhang, Guoxi Li

**Affiliations:** 1College of Animal Science and Technology, Henan Agricultural University, Zhengzhou 450001, China; 2College of Biological Engineering, Henan University of Technology, Zhengzhou 450001, China; 3Henan Key Laboratory for Innovation and Utilization of Chicken Germplasm Resources, Henan Agricultural University, Zhengzhou 450001, China

**Keywords:** circular RNAs, Gushi chickens, skeletal muscle, regulatory network, ceRNA

## Abstract

Circular RNAs (circRNAs) play a significant regulatory role during skeletal muscle development. To identify circRNAs during postnatal skeletal muscle development in chickens, we constructed 12 cDNA libraries from breast muscle tissues of Chinese Gushi chickens at 6, 14, 22, and 30 weeks and performed RNA sequencing. In total, 2112 circRNAs were identified, and among them 79.92% were derived from exons. CircRNAs are distributed on all chromosomes of chickens, especially chromosomes 1–9 and Z. Bioinformatics analysis showed that each circRNA had an average of 38 miRNA binding sites, 61.32% of which have internal ribosomal entry site (IRES) elements. Furthermore, in total 543 differentially expressed circRNAs (DE-circRNAs) were identified. Functional enrichment analysis revealed that DE-circRNAs source genes are engaged in biological processes and muscle development-related pathways; for example, cell differentiation, sarcomere, and myofibril formation, mTOR signaling pathway, and TGF-β signaling pathway, etc. We also established a competitive endogenous RNA (ceRNA) regulatory network associated with skeletal muscle development. The results in this report indicate that circRNAs can mediate the development of chicken skeletal muscle by means of a complex ceRNA network among circRNAs, miRNAs, genes, and pathways. The findings of this study might help increase the number of known circRNAs in skeletal muscle tissue and offer a worthwhile resource to further investigate the function of circRNAs in chicken skeletal muscle development.

## 1. Introduction

Skeletal muscle is an essential component of an animal’s body, and the growth and development of skeletal muscle has an influential effect on the meat production performance of poultry, which will directly determine the economic value of poultry [1]. One of the major sources of protein for humans is consuming chicken [2], and muscle development in chickens has been a major point of breeding investigation. Muscle development is a complicated multi-stage biological process, with many genes synergistically regulating each stage [3]. Thus, it is vital to reveal the molecular mechanisms of muscle development to promote skeletal muscle development and genetic breeding for meat quality.

CircRNA is a newly developed non-coding RNA transcript, which has a unique closed ring structure, which makes circRNAs more stable than linear RNA [4]. In recent years, due to the rapid development of high-throughput sequencing, thousands of circRNAs have been discovered in various organisms [5,6,7,8]. CircRNAs are widespread in various tissues and cells of a variety of species [9]. Research has suggested that circRNAs might be engaged in a variety of biological processes, affecting cellular physiological functions through various molecular mechanisms. CircRNAs can also be a target for miRNA- or RNA-binding proteins for regulating gene expression or translating proteins [10], which can play a key role in physiological activity and disease. Therefore, circRNAs have become a focus of research in the field of RNA in recent years. More and more studies have shown that circRNAs play an important role in eukaryotic organisms such as humans, chickens, and mice, and regulate the expression of muscle-related genes through a ceRNA mechanism [11,12]. For example, circSVIL promotes embryonic skeletal muscle development by sequestering miR-203 in chicken [13]. CircRNA_09505 can act as a miR-6089 sponge and regulate inflammation in CIA mice through the miR-6089/*AKT1*/NF-κB axis [14]. In addition, circRNAs can also regulate myogenesis through other regulatory mechanisms, and circZNF609 is translated into proteins that control the proliferation of myoblasts [15]. Currently, despite some progress in genetic breeding, there are few studies on the role of circRNAs in chicken skeletal muscle development. Thus, it is of great significance to explore the expression characteristics of circRNAs related to skeletal muscle development during chicken muscle development.

Local chicken breeds are important materials for poultry breeding and quality chicken production in China. Therefore, understanding the epigenetic regulation mechanism of the skeletal muscle development of local chickens will be beneficial to the conservation and exploitation of resources. Gushi chicken is an egg–meat type, which is mainly distributed in Gushi County, China. It is often used as a breeding and production material for its strong disease resistance and delicious meat quality. Gushi chicken has many good characteristics, but the growth rate is slightly slow. In order to learn and regulate the development of skeletal muscle of Gushi chickens, it is necessary to learn the molecular regulatory mechanisms of various developmental stages of skeletal muscle [16]. Our previous reports on breast muscle histology showed that muscle fiber diameter increases quickly before 22 weeks of age, while the relationship between breast fiber diameter and density remains balanced after 22 weeks of age [17]. In order to clarify the role of circRNAs in the development of skeletal muscle of Gushi chickens, we identified circRNAs by deep sequencing data at four stages of breast muscle development after birth (6, 14, 22, and 30 weeks). The circRNAs transcriptome profile of Gushi chickens and the circRNAs regulatory network associated with skeletal muscle development were constructed. These results might be used to explore further the molecular regulatory mechanisms of circRNAs in chicken skeletal muscle development and offer basic data for protecting Gushi chicken resources and mining useful traits.

## 2. Materials and Methods

### 2.1. Ethics Approval

The animal care in this research was performed according to the Regulations for the Management of Animal Experiments (Ministry of Science and Technology, Beijing, China, 2004) which were approved by the Animal Care and Use Committee of Henan Agricultural University, China.

### 2.2. Experimental Animals and Total RNA Extraction

The experimental animals were 200 Gushi chickens, which were transferred to cages after 4 weeks of age. The animal rearing and managing conditions were described as follows: before 14 weeks, the crude protein content of the diet was 18.5%, and the energy was 12.35 MJ/kg. After 14 weeks, the crude protein content of the diet was 15.6%, and the energy was 12.75 MJ/kg [17]. Three healthy animals were chosen randomly at four developmental stages at 6, 14, 22 and 30 weeks old, respectively. After dissection, the left breast muscle tissue was isolated and stored at −80 °C. Total RNA was isolated from breast muscle tissue using Trizol reagent (Takara, Dalian, China) following the manufacturer’s instructions, then a total of 12 RNA samples, 3 samples for each age group, were obtained to eliminate inter-individual differences. The quality of total RNA was assessed using 1% agarose gel electrophoresis and an Agilent Bioanalyzer 2100 system (Agilent, Santa Clara, CA, USA).

### 2.3. Library Construction and Sequence Analysis

A total of 5 μg of total RNA was taken from each sample as input for library construction, and rRNA was removed from total RNA using the Epicentre RIBO-ZERO™ kit (5602 Research Park Blvd., Suite 200 Madison, Epicentre Biotechnologies, Madison, WI, USA). The 250–300 bp short fragments were prepared as synthetic double-stranded cDNA; then A-tail was added, the sequencing adapter was connected, and the cDNA was obtained by PCR amplification. The libraries were diluted to 1 ng/μL, and the insert size of the library was checked with Agilent 2100 to make it meet the expected size; the library was quantified by quantitative PCR (qPCR) method to ensure an effective library level of the library was >2 nM. Finally, the obtained cDNA libraries were sequenced on the Illumina HiSeq2500 sequencing platform. Quality of the sequencing data was assessed by removing linker sequences and low mass sequences from the raw data and calculating the Q20, Q30 and GC content of the clean data. The clean data obtained were used for follow-up analysis. Alignment of paired clean reads to the reference genome used Bowtie software [18].

### 2.4. Identification of circRNAs

The circRNAs were identified by find_circ [19] and CIRI2 [20] software and the junctions of unaligned reads were extracted by reverse shearing. Each software uses the system as default for predictive analysis. Over 2 individual knots spanning reads and satisfying the Breathnach–Chambon rule (GU/AG rule) were determined as potential circRNAs. According to the location of circRNAs on the chromosome, the intersection of the two software results was taken as the identification of circRNAs in this report to reduce false positives. The circRNAs are fragments cut from different genes that are cyclized to form circRNAs, and these genes are called the source genes of circrNAs.

### 2.5. Differential Expression Analysis of circRNAs

Sequence reads for each circRNA in different libraries were counted based on the number of reverse splicing reads per million mappings, and corrected using TPM values. Normalized expression = (read count × 1,000,000)/libsize (libsize: sum of sample circRNA read count). Based on sequence readings, DESeq2 software [21] was applied to analyze the differential expression of six comparison combinations (W14 vs. W6, W22 vs. W14, W30 vs. W22, W22 vs. W6, W30 vs. W6, and W30 vs. W14) in circRNAs. The threshold criterion for judging the DE-circRNAs was *p* < 0.05.

### 2.6. Potential Functional Analysis of circRNAs

Functional enrichment analysis of all identified source genes of DE-circRNAs. Clusterprofiler package [22] was used for GO terms and KEGG pathway analysis of target genes, and presenting information by creating novel and high quality images with the R package ggplot2 [23]. Additionally, the significantly enriched GO terms and KEGG pathways were considered at calibrated *p*-value (*q*-value) < 0.05. The miRanda software was used to predict and analyze the miRNA target sites on the identified circRNAs to illustrate the interaction of circRNAs and miRNAs. IRESfinder software [24] was used to predict and identify IRES on circRNA sequences to identify circRNAs with translation potential for peptides or proteins. IRES is an essential element in the regulation of RNA translation that is independent of the 5′ cap structure. In circRNAs, IRES can regulate the ribosome size subunit assembly, enabling circRNAs to translate polypeptides or proteins.

### 2.7. Interaction Analysis of circRNA-miRNA-mRNA

Using the same tissue samples as the circRNA library, the mRNA (PRJNA516810) and miRNA (PRJNA516961) transcriptome profiles of 6-, 14-, 22-, and 30-week-old Gushi chickens’ breast muscles were constructed [16]. From these sequencing data, miRanda software [25] was used to predict the target genes of DE-circRNAs and their binding sites on miRNAs. Differentially expressed circRNA-miRNA-mRNA pairs were identified by Pearson correlation analysis according to the expression levels of circRNAs, miRNAs, and genes in the four developmental stages of Gushi chicken breast muscle. Using the DAVID online tool, GO and KEGG annotations were performed on all target genes in circRNA-miRNA-mRNA pairs. From the biological processes or pathways related to muscle development and cell growth, circRNA-miRNA-mRNA pairs with related relationships were selected, and Cytoscape [26] software (http://www.cytoscape.org/, accessed on 5 March 2022) was used to construct their interaction network.

### 2.8. Validation of circRNAs

First, the circRNA sequences from RNA-seq analysis were validated by real-time PCR (RT-PCR) analysis and DNA sequencing. Total RNA was extracted from breast muscle tissue of Gushi chickens of different ages, equal mixing, and used for cDNA synthesis. After RNA purification, cDNA was synthesized using RT-PCR kit (Vazyme, Nanjing, China). Eight circRNAs were randomly selected, and validation of adapter sequences was performed by PCR amplification using divergent primers (Appendix A). The PCR products were resolved by gel electrophoresis and DNA sequencing. The sequences obtained by DNA sequencing of PCR products were matched to chicken reference genome and RNA-seq data by using DNAMAN version 6.25 software [27] to verify the linker sequences of circRNAs.

Second, resistance of circRNAs to RNase R digestion was tested. The expression levels of the above 8 circRNAs were detected by quantitative real-time PCR (qRT-PCR) analysis. Prior to cDNA synthesis, total RNA was processed with RNase R (RNR-07250, epicenter). cDNA was synthetic from RNase R-treated RNA (RNase R^+^) and RNase R untreated RNA (RNase R^−^) by RT-PCR kit. qPCR was preformed using SYBR qPCR Master Mix (Vazyme, Nanjing, China). All the reactions were repeated three times. The relative expression levels of circRNAs were computed by the 2^−^^△△CT^ method [28], and the expression levels of the tested circRNAs were corrected using *β-actin* as an internal reference.

### 2.9. Data Analysis

Statistical analysis of qRT-PCR quantitative expression data was performed using GraphPad Prism 8.0 software (GraphPad software, Inc., San Diego, CA, USA). *t*-test [29] was performed to evaluate whether the data were statistically different (* *p* < 0.05, ** *p* < 0.01). Data are presented as the mean of three replicates.

## 3. Results

### 3.1. Overview of circRNA Library Sequencing and Identification

Twelve cDNA libraries were built using 6-, 14-, 22- and 30-week-old Gushi chicken breast muscle tissues. The number of raw reads obtained from each library varied from 89,496,872 to 117,197,064. After excluding adapter reads and low-quality reads, each library obtained between 12.75 G and 16.66 G of pure base data, representing more than 94%. The sequence export and quality assessment for each library is as follows (Appendix A). Using two pieces of software, find_circ2 and CIRC2, 2112 circRNAs were identified from 12 cDNA libraries in Gushi chicken breast muscle tissue after taking the intersection (Appendix A). Among them, 1831 circRNAs were sourced from 1162 known genes, and 281 circRNAs had no clear gene origin. The full-length range of these circRNAs was 166–98,282 nt, and the shear length range was 43–1148 nt. Among them, there were 651 circRNAs with a full length of more than 10,000 nt, 1249 circRNAs between 1000 and 9999 nt, and 212 circRNAs with a length of less than 1000 nt (Appendix A). These circRNAs were mainly sheared from the exon portion of the source gene, accounting for 79.92% on average, next to the intergenic regions (12.30%) and introns (7.78%) (Appendix A). These circRNAs are located on all chromosomes. The top 10 chromosomes with the largest number of circRNAs were chr1 (16.81%), chr2 (13.69%), chr3 (12.32%), chr4 (6.52%), chr7 (6.14%), chr5 (5.48%), chr6 (4.20%), chrZ (3.82%), chr8 (3.21%), and chr9 (2.46%) (Appendix A).

Eight circRNAs were selected randomly from the library and subjected to validation by PCR amplification and DNA sequencing. The results showed that the PCR amplification bands had bright bands at the anticipated size (Figure 1A); DNA sequencing confirmed that these circRNAs had head-to-tail connections (Figure 1B). In addition, the resistance test of circRNAs to RNase R digestion was further completed, and the results confirmed that all tested circRNAs were resistant to RNase R digestion (Figure 1C). All of these results indicate that the circRNAs identified in this research were authentic and credible.

### 3.2. Expression Characteristics of circRNAs

The identification of circRNA expression levels was normalized by TPM values. The spread of TPM values indicated a highly consistent circRNA expression profile with two peaks in the four developmental stages of the postnatal breast muscle in Gushi chickens. These circRNAs were primarily found in two ranges of TPM < 0.1 and TPM > 60. The proportion of their number was 21.26% and 73.03%, respectively. However, circRNA expression profiles differ in different developmental stages of the breast muscle of Gushi chickens, and there were many extremely highly expressed circRNAs in each developmental stage (Figure 2A,B).

In addition, a total of 543 significantly DE-circRNAs were identified from six comparison combinations (Appendix A). Among them, the DE-circRNAs in W14 vs. W6, W22 vs. W6, W22 vs. W14, W30 vs. W6, W30 vs. W14, and W30 vs. W22 combinations were 120 (up-regulated 47, down-regulated 73), 204 (up- 99, down- 105), 106 (up- 65, down- 41), 201 (up- 61, down- 140), 138 (up- 59, down- 79), and 81 (up- 25, down- 56), respectively (Figure 2C). Hierarchical cluster analysis indicated that the intra-group differences were less than the inter-group differences, proving the reliability of the data (Figure 2D). K-means cluster analysis showed that the expression modes of these DE-circRNA could be classified into four types. Among them, DE-circRNAs within subcluster_1 showed an overall down-regulation trend. DE-circRNAs in subcluster_2 showed an overall down-regulation trend, and the down-regulation at 14, 22 and 30 weeks was significantly lower than that of the breast muscle samples at 6 weeks. The DE-circRNAs in subcluster_3 showed an overall up-regulation trend, and the relative expressions showed an increasing trend at 6, 14 and 22 weeks, and then decreased after 30 weeks. DE-circRNAs in subcluster_4 showed an overall down-regulation trend, and the relative expressions were the lowest at 30 weeks of age, three of which showed obvious temporal characteristics with the development of the breast muscle of Gushi chickens (Figure 2E), suggesting that they may play an essential role in the developing of breast muscle. We also counted the distribution of common and unique DE-circRNAs in the six comparative combinations (Figure 2F), and found that there was no shared DE-circRNA among the six comparative combinations, and of any five comparative combinations, only gga_circ_0007220, gga_circ_0002213, gga_circ_0005531, gga_circ_0001397, and gga_circ_0000873 differential expression appeared in the four comparison groups at the same time. These results also indicated that circRNAs were clearly stage-specific within the development of the Gushi chicken breast muscle.

### 3.3. Potential Functions of DE-circRNAs in Chicken Breast Muscle Development

We completed the functional enrichment analysis of DE-circRNAs source genes. GO enrichment analysis showed that there were significant enrichments for terms such as cell development, differentiation, sarcomere, and myofibril formation (Figure 3A). In addition, the results of KEGG pathway analysis showed that the mTOR signaling pathway, TGF-β signaling pathway, and Wnt signaling pathway related to skeletal muscle development were significantly enriched (Figure 3B). The findings suggest that there are significant differences in the function of DE-circRNAs at different stages of breast muscle development in Gushi chickens. Therefore, we speculate that DE-circRNAs might be involved in the modulation of skeletal muscle development in Gushi chickens through these pathways and play an important role.

In addition, we focused on two pathways that were significantly enriched in mTOR signaling pathway and TGF-β signaling pathway. It is found that the elements in the pathway are the source genes of DE-circRNAs, which together form an interactive regulatory network (Figure 4). The mTOR signaling pathway and TGF-β signaling pathway and their enriched genes are known to be closely related to skeletal muscle development, such as mitogen-activated protein kinase 3 (*MAPK3*), phosphatase and tensin homolog (*PTEN*), etc. These results suggest that circRNAs may play an essential role in skeletal muscle development through the pathway of their source genes.

### 3.4. Analysis of miRNA Targeting Sites on circRNAs

Prediction of binding sites for known chicken miRNA on identified circRNAs using miRanda software. The results showed that there were a total of 80,055 miRNA-binding sites in the 2112 circRNAs identified, which could be bound by 1234 miRNAs. On average, there are 38 miRNA binding sites per circRNA, so one circRNA can bind multiple miRNAs. An miRNA has 1–7 binding sites on a specific circRNA, and can also target multiple circRNAs at the same time. Targeting relationship analysis between DE-circRNAs and miRNAs is helpful for further study of the functions of circRNAs (Appendix A). Based on the results of miRNA binding site analysis, the interaction between these skeletal-muscle-development-related miRNAs and DE-circRNAs was screened, and differentially expressed circRNA-miRNA pairs were identified, and their interaction network was constructed (Figure 5). Among them, miRNAs such as miR-206 and miR-148a-3p can have targeting relationships with multiple circRNAs. There are multiple miRNA binding sites on circRNAs such as gga_circ_0004993, gga_circ_0003686, and gga_circ_0001479. The results suggest that these circRNAs may play important roles in chicken skeletal muscle development.

### 3.5. Prediction of circRNA-miRNA-mRNA Interaction Network Related to Skeletal Muscle Development

Based on the whole-transcriptome sequencing data and DE-circRNAs of the breast muscle of Gushi chickens at 6, 14, 22, and 30 weeks of age, the circRNA-miRNA-gene association analysis was completed, and a total of 12,367 circRNA-miRNA-gene pairs were identified in the six comparative combinations (Appendix A). Using the DAVID online tool, 498 genes in these circRNA-miRNA-gene pairs were subjected to GO and KEGG annotation and enrichment analysis. Many genes were found to be closely related to biological processes or pathways related to skeletal muscle development (Appendix A). Combined with the results of enrichment analysis, we screened circRNA-miRNA-gene pairs related to skeletal muscle development and cell growth processes and constructed their interactive ceRNA network.

In particular, the circRNA-miRNA-gene network related to skeletal muscle development contains 54 circRNA-miRNA-gene pairs, consisting of 25 circRNAs, 23 miRNAs and 11 genes (Figure 6). In this network, there are miRNAs such as gga-miR-206, gga-miR-148a-3p, and gga-miR-181b-5p, which are associated with ankyrin repeat domain 1 (*ANKRD1*), ankyrin repeat domain 12 (*ANKRD12*), *MAFF*, *FOS*, and osteocalcin-like protein OC3 (*OC3*), and other genes have a targeting relationship.

In addition, in the circRNA-miRNA-gene network related to cell growth, gga-miR-148a-3p, gga-miR-200b-3p, gga-miR-206, and gga-miR-29b-3p target more than five circRNAs, respectively (Figure 7). Gga-miR-148a-3p associates with lysosomal organelle complex 1 subunit 5 (*BLOC1S5*), ankyrin repeat domain 1 (*ANKRD1*), and *YWHAG*. gga-miR-206 associates with phytate kinase 2 (*IP6K2*) and Cyclin C (*CCNC*) has an interaction relationship. *ANKRD1*, *YWHAG*, and *IP6K2* are targeted by eight, eight, and five miRNAs, respectively. They constitute a complex ceRNA network and are involved in the control of cell growth.

### 3.6. Internal Ribosome Entry Site (IRES) Prediction of circRNAs

In this study, we found that 1295 circRNAs had IRES elements on their sequences, accounting for 61.32% of the predicted and analyzed circRNAs. We counted the interval distribution of the prediction scores of these circRNAs (Appendix A). The top 10 circRNAs with predicted scores were gga_circ_0000696, gga_circ_0007520, gga_circ_0002731, gga_circ_0009195, gga_circ_0005534, gga_circ_0002673, gga_circ_0002572, gga_circ_0006933, gga_circ_000894, and gga_circ_000894. These results indicate that the vast majority of circRNAs expressed during chicken breast muscle development have the potential to encode polypeptides or proteins.

## 4. Discussion

Studies have shown that circRNAs can regulate gene expression through a variety of mechanisms and participate in the regulation of various biological processes, including skeletal muscle development [10]. It is known that circRNAs have expression specificity in different tissues, cells, and developmental stages [30]. Previous studies have shown that circRNAs have different expression patterns during fetal and postnatal developmental in goat skeletal muscle, and their chromosomal distribution is mainly on chromosome 1 [31]. Similarly, our results also showed that circRNAs expressed in skeletal muscle of chickens were mainly distributed on chromosome 1, which indicated that circRNAs on chromosome 1 played an important role in skeletal muscle development. Studies have shown that circRNAs are mainly found on chromosomes 1–10 and sex chromosomes. We speculate that one reason for this is because the longer the chromosome, the more circRNAs it contains. In nine tissues of pigs and three developmental stages of skeletal muscle, it has also been demonstrated that the expression of circRNAs has significant tissue specificity [6]. CircRNAs also had different expression patterns in the leg muscle of Xinghua chickens of various embryonic ages [32]. Similar to these results, the TPM values of the circRNAs identified in this report had consistent distribution patterns of the four developing stages of the breast muscle of Gushi chickens, which indicated that circRNAs had a particular expression mode in Gushi chicken breast muscle. In addition, the identified 543 DE-circRNAs had obvious temporal expression characteristics, and there were few differentially expressed circRNAs in all comparison groups, suggesting that the expressing of circRNAs in the breast muscle of Gushi chickens has a significant temporal specificity. In conclusion, circRNAs with specific expression characteristics may play an essential regulatory role in chicken skeletal muscle development.

Most of the source genes of circRNAs encode proteins, while circRNAs can affect the selective splicing of source genes and regulate the expression of source genes. Additionally, most circRNAs are spliced from exons, which mainly exist in the cytoplasm [33,34,35], meaning that that most of the circRNAs may express proteins. In this study, 1831 circRNAs were identified from 1162 known genes, most of which were derived from gene exons. We discovered that some source genes of circRNAs were firmly associated with the development of skeletal muscle. Cyclin T2 (*CCNT2*), the source gene of gga_circ_0009799, is closely related to the development of skeletal muscle [36], and gga_circ_0009799 is a significant differential circRNA for the three comparison groups of W14 vs. W6, W22 vs. W6, and W30 vs. W6, which implies that gga_circ_0009799 might have an important role in skeletal muscle development in Gushi chickens. *ACTA1*, the source gene of gga_circ_0006492, represses skeletal muscle satellite cell proliferation in pigs by downregulating the level of cell-cycle-related gene expression [37], and gga_circ_0006492 was a significantly different circRNA in our results for the two comparison groups W30 vs. W14, W30 vs. W22, suggesting that gga_circ_0006492 may play a key regulatory function in the later stages of skeletal muscle development in Gushi chickens. The source gene forkhead box P1 (*FOXP1*) of gga_circ_0000808 is engaged in skeletal muscle development as a transcriptional inhibitor of gene expression [38], and gga_circ_0000808 is a significant differential circRNA of W30 vs. W14, W30 vs. W6, which predicts that gga_circ_0000808 may be involved in the skeletal muscle development process in Gushi chickens. In addition, ankyrin repeat domain 1 (ANKRD1), which has a targeting relationship with multiple DE-circRNAs such as gga_circ_0000915, gga_circ_0003999, gga_circ_0002053, and gga_circ_0005968, exists in the nuclei and sarcomeres of cardiac and skeletal muscle and is involved in connecting myofibril stress and plays a role in transcriptional regulation [39]. We also completed the functional enrichment analysis of DE-circRNAs in the breast muscle of Gushi chickens and found many GO terms and KEGG pathways related to skeletal muscle development, such as myofibril assembly, skeletal system development, cell differentiation, mTOR signaling pathway, TGF-β signaling pathway, etc. [40,41]. These results suggest that most circRNAs identified in this study may affect skeletal muscle development of Gushi chickens through the expression of their source genes. Further research can confirm the interaction between circRNAs and their source genes.

More and more evidence suggests that circRNAs can act as miRNA sponges to inhibit their activity, thereby regulating the expression and function of target genes [42]. Therefore, this study predicted the miRNA binding sites on circRNA sequences and found that most of the identified circRNAs had multiple potential miRNA binding sites. These results are consistent with previous studies [8,16,32]. This suggests that circRNAs can play a regulatory role in skeletal muscle development by binding to miRNAs. Therefore, we constructed circRNA-miRNA networks and circRNA-miRNA-gene pairs using miRNAs that have been previously shown to be associated with skeletal muscle development. In the network, circRNAs sequences such as gga_circ_0000385, gga_circ_0001431, gga_circ_0001685, gga_circ_0002605, and gga_circ_0003686 all have five or more miRNA binding sites. Among these miRNAs that can bind to circRNAs, miR-206 is an essential regulator of skeletal myoblast differentiation [43,44]. MiR-148a-3p contributes to myoblastic differentiation by targeting *DYNLL2* and promoting myosin heavy chain (MYHC) protein expression [45]. MiR-499-5p can regulate the development of skeletal muscle by regulating muscle gene expression through the NFATc1/MEF2C pathway [46]. In addition, studies have found that circRNA FUT10 can act as a sponge for miR-365a-3p to regulate the expression of *HOXA9* and then regulate the senescence of skeletal muscle stem cells [47]; a novel circular RNA produced by *FGFR2* gene by sponge miR-133a-5p and miR-29b-1-5p promotes myoblast proliferation and differentiation [12]; circMYBPC1 can promote myoblast differentiation by targeting *MyHC* and may promote skeletal muscle regeneration [48]. These findings suggest that circRNAs may play an intrinsic competition-regulated role during skeletal muscle development by interacting with miRNA. The specific regulatory mechanism of circRNAs as miRNA molecular sponges in chicken skeletal muscle development needs to be further studied.

Skeletal muscle development is an extremely complex process involving muscle formation, myogenic fiber assembly and development, myogenic cell proliferation and differentiation, intercellular connectivity and other biological processes, regulated by a variety of signaling pathways [49,50]. CircRNA is known to regulate the function of miRNA through the mechanism of ceRNA [51,52], which in turn affects the regulation of miRNAs on downstream genes [53]. Research has shown that circRNAs can affect biological processes such as cell proliferation by affecting signaling pathways [10,54]. Therefore, we constructed ceRNA networks related to muscle development and cell growth processes based on the results of association analysis and pathway enrichment. In these circRNA-miRNA-gene interaction networks, complex relationships were formed between circRNAs and miRNAs, miRNAs and their target genes, as well as between various pathways, which jointly regulate skeletal muscle development in Gushi chickens. For example, the mTOR signaling pathway and TGF-β signaling pathway have both been shown to be involved in the regulation of skeletal muscle growth [41,55]. *ANKRD1* is present in the nuclei and sarcomeres of cardiac and skeletal muscle, plays a role in linking myofibril stress and transcriptional regulation, and is involved in muscle tissue adaptation and remodeling [39]. Variability in the *FOS* gene is one of the factors affecting porcine skeletal muscle fiber and metabolic characteristics [56]. These results suggest that complex interactions between genes and pathways may exist during skeletal muscle development.

Recently, circRNAs were found to have IRES, which can bind promoters or their own ribosomes to drive ORF translocation and have the potential for directly encoding proteins or functional small peptides [57,58,59]. Studies have shown that circ-ZNF609 is associated with heavy polyribosomes and is translated into proteins in splicing and cap-independent ways to regulate the proliferation of myoblasts [15]. In this study, we predicted that 1295 circRNAs contained IRES elements, suggesting that many circRNAs may have the potential to encode proteins or polymorphisms during chicken skeletal muscle development. The role of IRES of these circRNAs in skeletal muscle development remains to be further elucidated in terms of proteins or peptides.

## 5. Conclusions

This study describes the circRNA profiles of Gushi chicken breast muscle at 6, 14, 22, and 30 weeks. Many circRNAs and DE-circRNAs associated with chicken skeletal muscle development were identified, and a ceRNA regulatory network related to skeletal muscle development and cell growth was constructed. These results offer new insights and a valuable resource for better comprehending the regulation role of circRNAs in postnatal skeletal muscle development in chickens.

## Figures and Tables

**Figure 1 genes-13-01974-f001:**
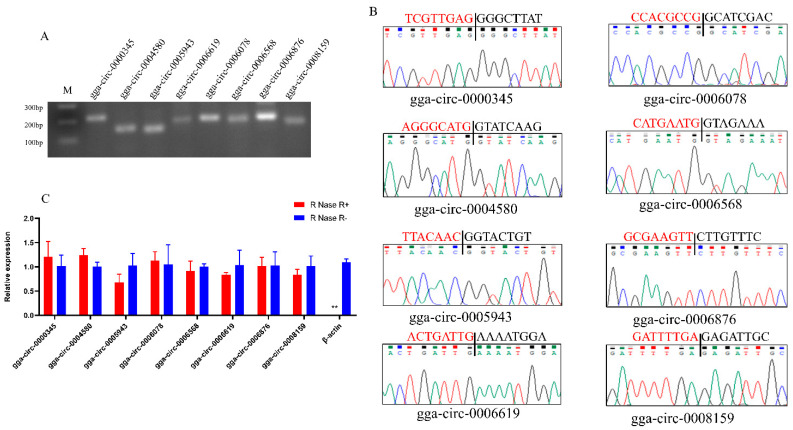
Validation of eight circRNAs. (**A**) Electropherogram of PCR products. M: marker. (**B**) Sanger sequencing confirmed the head-to-tail junction of eight circRNAs. Black lines indicate the location of back splicing. The region marked in red (or black) is the terminal (or start) sequence of the circRNA (junction point). (**C**) Resistance of circRNAs to RNase digestion was detected by qRT-PCR. Control was *β-actin*. Data are presented as mean ± SEM (n = 3).

**Figure 2 genes-13-01974-f002:**
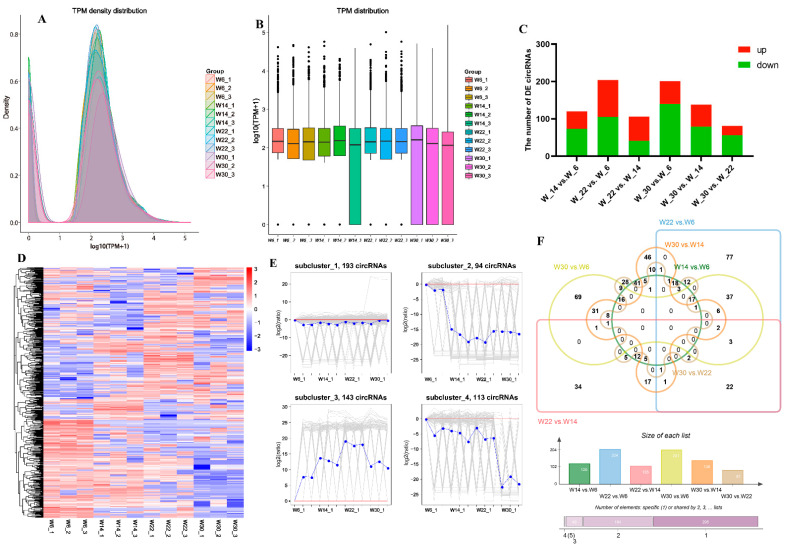
Expression analysis of circRNAs. (**A**) Density distributions of circRNAs TPM values in various libraries. (**B**) Boxplots of TPM values of circRNAs in various libraries. (**C**) DE-circRNAs in six comparison groups. The abscissa represents different comparison groups, and the ordinate represents the number. Green bars represent the number of down-regulated DE-circRNAs, while red bars represent the number of up-regulated DE-circRNAs. (**D**) Hierarchical clustering of DE-circRNAs. (**E**) K-means cluster plot of DE-circRNAs. (**F**) Venn diagram of the DE-circRNAs in different comparison groups.

**Figure 3 genes-13-01974-f003:**
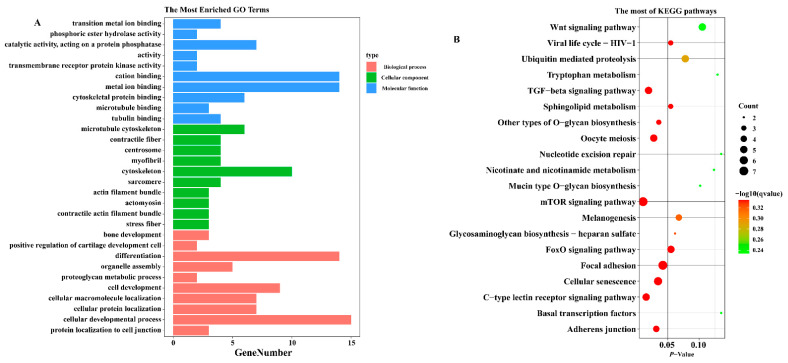
GO and KEGG functional enrichment analysis of source genes of DE-circRNAs. (**A**) The *y*-axis is the GO term at the next level of the three GO categories, and the *x*-axis is the number of source genes annotated for that term. (**B**) The *y*-axis indicates the pathway name, the *x*-axis indicates the *p*-value, the size of the dots indicates the number of source genes in the pathway, and the color of the dots corresponds to the different −log10 (*q*-value) ranges.

**Figure 4 genes-13-01974-f004:**
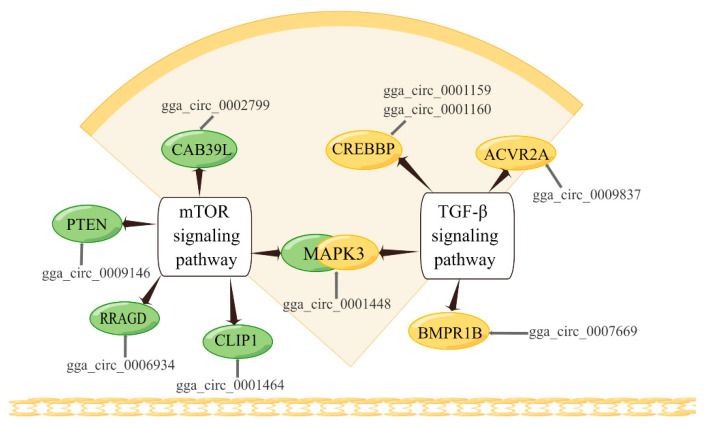
Skeletal-muscle-development-related pathways involved in differentially expressed mRNAs and DE-circRNAs.

**Figure 5 genes-13-01974-f005:**
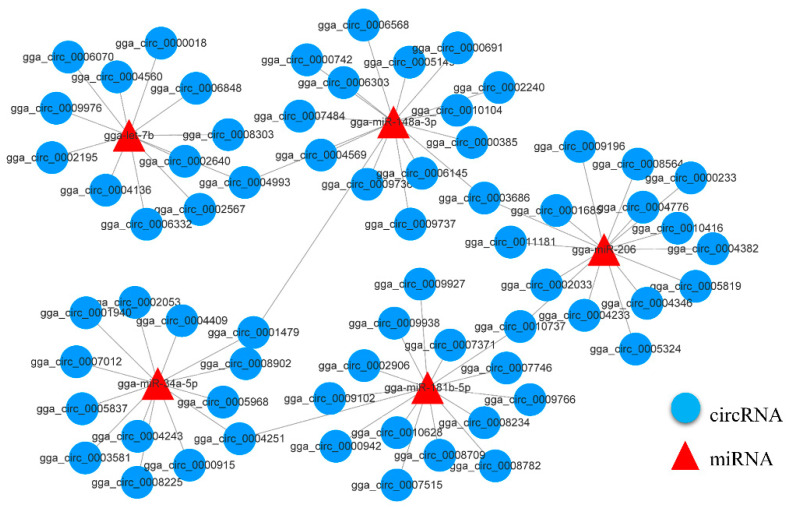
CircRNA-miRNA interaction network associated with skeletal muscle development. miRNAs and CircRNAs are expressed as triangles and circles, respectively.

**Figure 6 genes-13-01974-f006:**
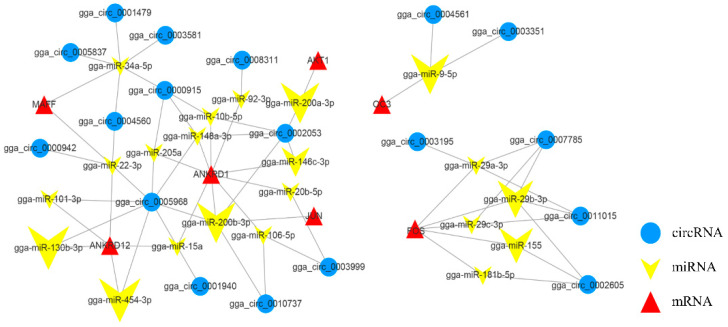
CircRNA-miRNA-gene interaction network related to skeletal muscle development. The circle, inverted arrows, and triangles represent circRNAs, miRNAs, and mRNAs, respectively.

**Figure 7 genes-13-01974-f007:**
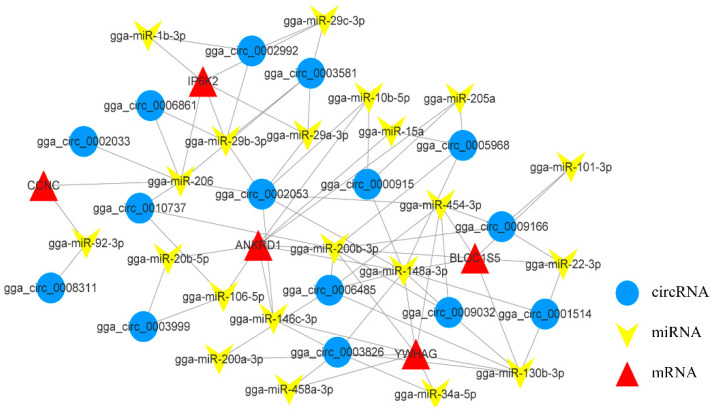
CircRNA-miRNA-gene interaction network related to cell growth. The circle, inverted arrows, and triangles represent circRNAs, miRNAs, and mRNAs, respectively.

## Data Availability

The datasets used and analyzed during the current study are available from the corresponding author on reasonable request. Transcriptome data is deposited in the NCBI database under the sequence read archive with accession numbers PRJNA516961 (miRNA) and PRJNA516810 (mRNA).

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
