# Peer review of "CircRNAs Related to Breast Muscle Development and Their Interaction Regulatory Network in Gushi Chicken"

_genes, 2022, doi:10.3390/genes13111974_

Round 1

Reviewer 1 Report

I reviewed the manuscript entitled “CircRNAs related to breast muscle development and their interaction regulatory network in Gushi chicken” for publication in Genes as an original article. The author investigated circular RNAs of breast muscle tissue at 6, 14, 22, and 30 weeks to know how circRNAs influence skeletal muscle development in Gushi chickens. Although the sample size was minimum, the results looked fine and included novel findings. I found several points that should be revised. 

My comments will be shown below:

[1]. The authors used 200 Gushi chickens at 4 weeks of age. However, only 12 individuals were used for analyzing circRNAs. I am wondering performance of the remaining 188 individuals. I think 3 individuals will be the minimum sample size in each stage in the first screening. How about using additional 10 or more individuals for the validation experiment? The authors should discuss the sample size.

[2]. In Figure 2, six figures (A-F) were shown in the small space. It is hard to see the letters in the x and y axis and legends because of the tiny letters. The authors should indicate bigger figures.

[3]. Figure 3 has the same problem stated above. Almost all letters cannot recognize. Please revise them.

[4]. In Figure 4, which are the differentially expressed circRNAs? The authors should indicate the legend of the figure.

Author Response

Dear reviewers,

Thank you very much for your kindly comments on our manuscript. There is no doubt that these comments are valuable and very helpful for revising and improving our manuscript. In what follows, we would like to answer the questions you mentioned and give detailed account of the changes made to the original manuscript.

[1]. The authors used 200 Gushi chickens at 4 weeks of age. However, only 12 individuals were used for analyzing circRNAs. I am wondering performance of the remaining 188 individuals. I think 3 individuals will be the minimum sample size in each stage in the first screening. How about using additional 10 or more individuals for the validation experiment? The authors should discuss the sample size.

Response: Thanks for your valuable counsel. In this study, we reared 200 Gushi chickens in the same environment with the standard conditions for pure breeding and conservation of Gushi chickens, and randomly selected three healthy chickens at 6 weeks, 14 weeks, 22 weeks and 30 weeks of age, respectively, we thought that the individuals we selected could represent the characteristics of the population, and in addition, considering the cost, only three were selected for sequencing and validation at that time, and the rest continued to be used as pure breeding of Gushi chickens.

[2]. In Figure 2, six figures (A-F) were shown in the small space. It is hard to see the letters in the x and y axis and legends because of the tiny letters. The authors should indicate bigger figures.

Response: Thank you for your comments on our manuscript. We have adjusted the font in the image and displayed the image in a higher resolution, please see Figure 2.

[3]. Figure 3 has the same problem stated above. Almost all letters cannot recognize. Please revise them.

Response: Thank you for pointing this out. We have adjusted the font in the image and displayed the image in a higher resolution, please see Figure 3.

[4]. In Figure 4, which are the differentially expressed circRNAs? The authors should indicate the legend of the figure.

Response: Thank you for your significant reminding. In Figure 4 we show all DE-circRNAs, which have been annotated in the manuscript with modifications and marked in red font, see lines 273-275 of the manuscript.

We tried our best to improve the manuscript and made some changes marked in red in revised paper which will not influence the content and framework of the paper. We would like also to thank you for allowing us to resubmit a revised copy of the manuscript.

Best regards,

Mr. Yuan

Reviewer 2 Report

The authors have profiled Gushi chicken breast muscle transcriptome and have identified CircRNAs and concluded that several of the identified CircRNAs play a role in skeletal muscle development.

 Was the dataset used in this study the same as that of “LncRNAs and their regulatory networks in breast muscle tissue of Chinese Gushi chickens during late postnatal development” BMC genomics 2021. “MicroRNAs and their regulatory networks in Chinese Gushi chicken abdominal adipose tissue during postnatal late development” BMC genomics 2019. “Analyses of MicroRNA and mRNA Expression Profiles Reveal the Crucial Interaction Networks and Pathways for Regulation of Chicken Breast Muscle Development” Front in genetics 2019. If so, proper attributions to the same must be given including citation.

In your “Characteristics and expression profiles of circRNAs during abdominal adipose tissue development in Chinese Gushi chickens” paper in Plos one 2021. You had reported that the majority of CircRNAs were within GGA 1 – 10 and Z chromosome, you have reported the same here too, did you use the same CircRNAs identified in the earlier study ? Or this could just be a function of the chromosome being larger than other chromosomes.  (lines 200 – 202)

From your description of experimental animals and RNA library preparations and comparison to the three other paper on Gushi chicken, it seems that these are data from the same animals and same libraries, if so, why does it warrant separate papers?  Can’t it all be part of the same paper.

Line 139 :- What were the inputs for miRanda software, did you download miRNAs or were they from your previous work ?

In some places you have mentioned Source gene, while in some place’s, parental gene, what do you mean by these terms and how were these source/parental gene identified. (Line 133)

Line 146 -148:  Are these from the two papers mentioned earlier? if so cite them.

On line 197 “The main source genes of these circRNAs were exons “ What does this mean? Please clarify this statement and indicate what this could mean.

Figure 2 is illegible and impossible to read. Please provide higher quality figures.

Line 234-236 :- What are the roles or CircRNAs or their source genes/ miRNAs in the different clusters ?

Figure 3 is hard to read. In panel B, the legend says -log10 (qvalue), while the values are in decimals and the axis label on x axis says p-value. Its not clear what values were plotted other than gene count (size of the dots) for each enriched pathway. Please clarify.

You identified DE CircRNAs for each stage (W14 vs W 6 etc), did you group all the DE CircRNAs and perform the functional enrichment analysis? Can you provide a rationale for doing so?

Line 291 – you have identified miRNA targets on DE circRNAs, what lead you to conclude these 3 CircRNAs to play important role in chicken skeletal muscle development. This statement is without any evidence? You should likely identify the target genes for these miRNAs and see what functions those genes have.

Line 296 and below – Did you use DE CircRNAs or just all 2000 odd identified circRNAs ?

You have briefly mentioned how you performed circRNA-miRNA-gene association in lines 145-152. However its not clear. Did you identify DE miRNAs and then identified their target genes with miRanda or did you identify DE circRNA, then their target miRNAs using miRANDA and then their target mRNAs and then identify if these targets are DE in the different skeletal muscle stages sampled? and then use these for Pearson correlation?

The different genes, circRNAs mentioned in the discussion, were they DE. If so at what stage. These are necessary as you state in introduction (Line 73) that “understanding molecular regulation mechanisms of skeletal muscle at different stages is necessary “  and is a primary objective of your study.

Please provide a list of identified circRNA (Identifies given in your study, and their location in the genome, and whether they were DE at any stage in your study) without that it will not be useful for other researchers.

Author Response

Dear Reviewer:

Thank you for your decision and constructive comments on my manuscript. We have carefully considered your suggestions and made some changes. We have tried our best to improve and make some changes to the manuscript. Based on your comments, the revised parts have been marked in red font.

  1. Was the dataset used in this study the same as that of “LncRNAs and their regulatory networks in breast muscle tissue of Chinese Gushi chickens during late postnatal development” BMC genomics 2021. “MicroRNAs and their regulatory networks in Chinese Gushi chicken abdominal adipose tissue during postnatal late development” BMC genomics 2019. “Analyses of MicroRNA and mRNA Expression Profiles Reveal the Crucial Interaction Networks and Pathways for Regulation of Chicken Breast Muscle Development” Front in genetics 2019.If so, proper attributions to the same must be given including citation.

Response: Thank you very much for your comments on our manuscript. The sequencing data of circRNAs in this article are the same as the database in the “LncRNAs and their regulatory networks in breast muscle tissue of Chinese Gushi chickens during late postnatal development” article, and we have used the data from the “Analyses of MicroRNA and mRNA Expression Profiles Reveal the Crucial Interaction Networks and Pathways for Regulation of Chicken Breast Muscle Development”  article in the association analysis, which we have included in the article with explanations and citations marked in red, see lines 145-147 in the manuscript.

  1. In your “Characteristics and expression profiles of circRNAs during abdominal adipose tissue development in Chinese Gushi chickens” paper in Plos one 2021. You had reported that the majority of CircRNAs were within GGA 1 – 10 and Z chromosome, you have reported the same here too, did you use the same CircRNAs identified in the earlier study? Or this could just be a function of the chromosome being larger than other chromosomes.  (lines 200 – 202)

Response: Thanks for your valuable counsel. The circRNAs we reported previously in “Characteristics and expression profiles of circRNAs during abdominal adipose tissue development in Chinese Gushi chickens” paper in Plos one 2021. were mainly on GGA1-10 and sex chromosomes, and although the results are consistent, there are differences between the results reported in this paper and those reported previously because they are not from the same tissue samples. And in Circular RNAs are abundant and dynamically expressed during embryonic muscle development in chickens” paper in DNA Research 2018. also reported that in chick embryonic muscle circRNAs are predominantly on chromosomes 1-10 and sex chromosomes. These suggest that circRNAs may be predominantly on chromosomes 1-10 and sex chromosomes. We believe that one of the reasons for this is indeed because the longer the chromosome, the more circRNAs it contains.

  1. From your description of experimental animals and RNA library preparations and comparison to the three other paper on Gushi chicken, it seems that these are data from the same animals and same libraries, if so, why does it warrant separate papers?  Can’t it all be part of the same paper.

Response: Thank you very much for your comments on our manuscript. We have the same samples as the two previously published articles, but the data of circRNA and miRNA were built separately, because the focus of our research in each article is different and we also want to introduce the properties of different non-coding RNAs in Gushi chicken more clearly, so we consider to analyze and present them separately.

  1. Line 139: What were the inputs for miRanda software, did you download miRNAs or were they from your previous work?

Response: Thank you for pointing this out. We performed miRNA binding site analysis of circRNAs using miRanda software inputting all circRNAs identified in this study and all known chicken miRNAs that have been modified and marked in red in the manuscript, see lines 137-139, and all binding site information in Supplementary Table S7.

  1. In some places you have mentioned Source gene, while in some places, parental gene, what do you mean by these terms and how were these source/parental gene identified. (Line 133)

Response: Thank you for pointing this out. Here we want to express the meaning of the source gene, which has been corrected and marked in red in the manuscript, see line 267.

  1. Line 146 -148:  Are these from the two papers mentioned earlier? if so cite them.

Response: Thank you very much for your suggestions. Here we use the data part of the correlation analysis, which has been cited and marked in red, see lines 145-147 of the manuscript.

  1. On line 197 “The main source genes of these circRNAs were exons “What does this mean? Please clarify this statement and indicate what this could mean.

Response: Thank you very much for your comments on our manuscript. What we want to express here is that most circRNAs are mainly sheared from the exon portion of their source genes, which are known to be part of eukaryotes and can eventually be expressed as proteins, which means that probably most circRNAs may express proteins. Already revised and marked in red in the manuscript, see lines 195, 366-367.

  1. Figure 2 is illegible and impossible to read. Please provide higher quality figures.

Response: Thank you for pointing this out. We have adjusted the font in the image and displayed the image in a higher resolution, please see Figure 2.

  1. Line 234-236: What are the roles or CircRNAs or their source genes/ miRNAs in the different clusters?

Response: Thank you very much for your comments on our manuscript. Here we show the K-means cluster in Figure 2E, where the red horizontal line is used as a reference, the upper line is up-regulated and the lower line is down-regulated. Where DE-circRNAs within subcluster_1 showed an overall down-regulation trend. DE-circRNAs in subcluster_2 showed an overall down-regulation trend, and the down-regulation at 14, 22 and 30 weeks was significantly lower than that of the pectoral muscle samples at 6 weeks. DE-circRNAs in subcluster_3 showed an overall up-regulation The DE-circRNAs in subcluster_4 showed an overall down-regulation trend, and the relative expressions at 6, 14 and 22 weeks showed an increasing trend, and then decreased after 30 weeks. subcluster_4 showed an overall down-regulation trend, and the relative expressions reached the lowest at 30 weeks of age. These results suggest that DE-circRNAs may play a regulatory role in the later stages of pectoral muscle development in Gushi chickens. We have revised and marked in red font in the manuscript, see lines 231-237.

  1. Figure 3 is hard to read. In panel B, the legend says -log10 (qvalue), while the values are in decimals and the axis label on x axis says p-value. Its not clear what values were plotted other than gene count (size of the dots) for each enriched pathway. Please clarify.

Response: Thank you for your significant reminding. We have explained and marked in red in the example section of Figure 3, see lines 267-271 of the manuscript.

  1. You identified DE CircRNAs for each stage (W14 vs W 6 etc), did you group all the DE CircRNAs and perform the functional enrichment analysis? Can you provide a rationale for doing so?

Response: Thank you very much for your comments on our manuscript. We identified DE-circRNAs between different comparison groups, see Supplementary Table S4. we performed functional enrichment analysis of the source genes of DE-circRNAs from different comparison groups, see Supplementary Figure 4 and Supplementary Figure 5. due to the small number of DE-circRNAs between each comparison group, after functional enrichment, fewer pathways were significantly enriched in the results, again Considering that the later association analysis was done with all DE-circRNAs, the analysis was done using all DE-circRNAs of the source genes.

  1. Line 291 – you have identified miRNA targets on DE circRNAs, what lead you to conclude these 3 CircRNAs to play important role in chicken skeletal muscle development. This statement is without any evidence? You should likely identify the target genes for these miRNAs and see what functions those genes have.

Response: Thank you very much for your suggestions. During our analysis, we found that some circRNAs can exist binding sites with multiple miRNAs, and the reported miRNAs related to muscle development were also found, so we selected some of the results with targeting relationships to show them in the form of pictures, and these 3 circRNAs shown in the manuscript are some of them. And SVIL, the gene of source of gga_circ_0004993 mentioned in the paper, has been shown to encode a muscle-specific isoform of super vimentin (SV2), a large myocardin II- and f -actin-binding protein,Loss of supervillin causes myopathy with myofibrillar disorganization and autophagic vacuoles” paper in Brain 2021. The gene of source of gga_circ_0003686 mentioned in the paper Vascular hemophilia factor (VWF) is a plasma glycoprotein that regulates the proliferation of vascular smooth muscle cells. “The VWF/LRP4/αVβ3-axis represents a novel pathway regulating proliferation of human vascular smooth muscle cells.” paper in Cardiovasc Res. 2022. Myosin-18b (Myo18b), the gene of source of gga_circ_0001479 mentioned in the paper, has a targeting relationship with miR-760 to regulate the proliferation and differentiation of C2C12 myogenic cells. Taken together, these circRNAs may be involved in or regulate the process of chicken muscle development.

  1. Line 296 and below – Did you use DE CircRNAs or just all 2000 odd identified circRNAs?

Response: Thank you for your significant reminding. Here in the association analysis section we use the identified DE-circRNAs. have been modified in the manuscript and marked in red, see lines 304-305.

  1. You have briefly mentioned how you performed circRNA-miRNA-gene association in lines 145-152. However its not clear. Did you identify DE miRNAs and then identified their target genes with miRanda or did you identify DE circRNA, then their target miRNAs using miRANDA and then their target mRNAs and then identify if these targets are DE in the different skeletal muscle stages sampled? and then use these for Pearson correlation?

Response: Thank you very much for your comments on our manuscript. We are combining the pre-transcriptome data PRJNA516810 (mRNA) and small RNA library data PRJNA516961 (miRNA) for association analysis, using these data for association analysis of DE-circRNAs. miRanda software was used to predict the targeting interactions between DE-circRNAs and miRNAs. Differentially expressed circRNA-miRNA-gene pairs were identified by Pearson correlation analysis. We have made the corresponding changes in the manuscript and marked them in red, see lines 147-151.

  1. The different genes, circRNAs mentioned in the discussion, were they DE. If so at what stage. These are necessary as you state in introduction (Line 73) that “understanding molecular regulation mechanisms of skeletal muscle at different stages is necessary “and is a primary objective of your study.

Response: Thank you very much for your suggestions. We have revised and marked in red the corresponding DE-circRNAs in the discussion, see lines 370-374,376-379, 381-387 of the manuscript.

  1. Please provide a list of identified circRNA (Identifies given in your study, and their location in the genome, and whether they were DE at any stage in your study) without that it will not be useful for other researchers.

Response: Thank you very much for your comments on our manuscript. We provide this information in the Supplementary Table. A list of identified circRNAs with information is provided in Supplementary Table S3, and a list of DE-circRNAs in different comparison groups is provided in Supplementary Table S4.

We tried our best to improve the manuscript and made some changes marked in red in revised paper which will not influence the content and framework of the paper. We would like also to thank you for allowing us to resubmit a revised copy of the manuscript.

Best regards,

Mr. Yuan

Round 2

Reviewer 2 Report

Thank you for making the suggested changes and for your response to my comments. I have a few more suggestions/ comments

1) I suggest that you mention in the paper the large number of cirRNAs on GGA 1-10 and sex chromosome might be because they were the larger chromosomes.

2) The first mention of source gene is on line 133. You have to explain what you mean by source gene prior to this. So i suggest you add a line may be in section 2.4 about what you mean by source gene.

3) On Figure 3- in the legend , just use the terms x axis (for abscissa) and y axis (for ordinate) to describe the vertical and horizontal axis on figure 3A.

Author Response

Reply to Reviewer

Dear reviewers,

Thank you very much for your kindly comments on our manuscript. There is no doubt that these comments are valuable and very helpful for revising and improving our manuscript. In what follows, we would like to answer the questions you mentioned and give detailed account of the changes made to the original manuscript.

1) I suggest that you mention in the paper the large number of cirRNAs on GGA 1-10 and sex chromosome might be because they were the larger chromosomes.

Response: Thank you very much for your suggestion. We have added this section to the discussion section of the manuscript and marked the revisions with a yellow background. See lines 352-355.

2) The first mention of source gene is on line 133. You have to explain what you mean by source gene prior to this. So i suggest you add a line may be in section 2.4 about what you mean by source gene.

Response: Thank you for your comments on our manuscript. We have explained the source genes in the manuscript and marked the modified parts with a yellow background. See lines 123-124.

3) On Figure 3- in the legend, just use the terms x axis (for abscissa) and y axis (for ordinate) to describe the vertical and horizontal axis on figure 3A.

Response: Thank you for pointing this out. We have modified the legend section in Figure 3 and marked the modified section with a yellow background. See lines 268-270.

We have tried our best to improve the manuscript by making some changes to the yellow background markings in the revised paper, but they will not affect the content or framework of the paper. We would like also to thank you for allowing us to resubmit a revised copy of the manuscript.

Best regards,

Mr. Yuan
